

# Complex fault system revealed from 3-D seismic reflection data with deep learning and fault network analysis

Thilo Wrona[1,2*], Indranil Pan[3,4,5], Rebecca E. Bell[6], Christopher A-L. Jackson[7], Robert L. Gawthorpe[1], Haakon Fossen[8], Edoseghe E. Osagiede[1], and Sascha Brune[2,9]

[1]Department of Earth Science, University of Bergen, Allégaten 41, N-5007 Bergen, Norway.
[2]GFZ German Research Centre for Geosciences, Telegrafenberg, 14473 Potsdam, Germany.
[3]Centre for Process Systems Engineering & Centre for Environmental Policy, Imperial College London, UK.
[4]The Alan Turing Institute, British Library, London, UK.
[5]School of Mathematics, Statistics & Physics, Newcastle University, UK.
[6]Basins Research Group (BRG), Department of Earth Science and Engineering, Imperial College, Prince Consort Road, London, SW7 2BP, UK.
[7]Department of Earth and Environmental Sciences, University of Manchester, Manchester, UK
[8]Museum of Natural History, University of Bergen, Allégaten 41, N-5007 Bergen, Norway.
[9]Institute of Geosciences, University of Potsdam, Potsdam-Golm, Germany.

*Correspondence to*: Thilo Wrona (thilowrona@gmail.com)

**Abstract.** Understanding where normal faults are is critical to an accurate assessment of seismic hazard, the successful exploration for and production of natural (including low-carbon) resources, and for the safe subsurface storage of CO2. Our current knowledge of normal fault systems is largely derived from seismic reflection data imaging intra-continental rifts and continental margins. However, exploitation of these data is limited by interpretation biases, data coverage and resolution, restricting our understanding of fault systems. Applying supervised deep learning to one of the largest offshore 3-D seismic reflection data sets from the northern North Sea allows us to image the complexity of the rift-related fault system. The derived fault score volume allows us to extract almost 8000 individual normal faults of different geometries, which together form an intricate network characterised by a multitude of splays, junctions and intersections. Combining tools from deep learning, computer vision and network analysis allows us to map and analyse the fault system in great detail and a fraction of the time required by conventional interpretation methods. As such, this study shows how we can efficiently identify and analyse fault systems in increasingly large 3-D seismic data sets.

## 1 Introduction

Understanding the geometry and growth of normal fault systems is critical when assessing seismic hazard, when identifying suitable sites for subsurface $CO_2$ storage and when exploring for natural resources (traditional and low-carbon). For example, whereas probabilistic seismic hazard analyses based on seismic event catalogues are extremely useful when trying to forecast earthquake likelihood and location, high-resolution fault mapping, preferably in 3-D, can help us constrain the slip tendency of faults, where seismic catalogues are discontinuous and/or incomplete (e.g. Morris et al., 1996; Moeck et al., 2009;



Yukutake et al., 2015). Moreover, faults can facilitate (or impede) fluid and gas migration to the Earth's surface, thus
determining their geometry and connectivity, as well as their hydraulic properties is key for assessing their role in the long-
term subsurface storage of $CO_2$ (Bissell et al., 2011; Kampman et al., 2014). In both of these examples, we need robust
predictions of 3-D fault geometry over large areas and across a wide range of scales (100s m to 100 km).

Accurately mapping fault systems in 2-D and 3-D seismic reflection data typically requires expertise and time (e.g.
Bond, 2015). While we can map fault systems in great detail over small areas using 3-D seismic reflection data (e.g. Lohr et
al., 2008; Wrona et al., 2017; Claringbould et al., 2020), we lack an understanding of the character of 3-D fault populations
at the scale of entire rift systems, as regional studies are often limited to only sparse, 2-D seismic sections (e.g. Clerc et al.,
2015; Fazlikhani et al., 2017; Phillips et al., 2019). 3-D numerical models are now capable of simulating fault networks at
the rift scale; however, there are few observational data sets of the same scale to test the predictions of these models and,
therefore, help refine them (e.g. Naliboff et al., 2020; Pan et al., 2021).

Supervised deep learning allows us to map faults in seismic reflection data (e.g. Wu et al., 2019; Mosser et al.,
2020; Wrona et al., 2021a), but up until now these studies have been in the "proof of concept" phase, simply focusing on
detecting faults; such studies have yet to provide new insights into the geometry of normal faults. In this study, by applying
supervised deep learning to newly-acquired broadband 3-D seismic reflection data imaging much of the northern North Sea
rift (161 km wide in E-W, 266 km long area in N-S, 0-20 km deep), we map the fault network associated with a continental
rift basin at an unprecedented level of detail. Using manually labelled data (<0.1% of data volume), we train a deep
convolutional neural network (U-Net) to predict faults in our data set. The predicted score ranges from 0 (no fault) to 1
(fault). Based on this score across the entire 3-D seismic volume we employ a second workflow to extract the normal fault
system as a network (a set of nodes and edges) allowing us to investigate the architecture and growth of this extremely
complex system consisting of thousands of intersecting faults.

## 2 Geological setting

The study area is located in the northern North Sea (Fig. 1), where continental crust consists of 10–30-km-thick crystalline
basement overlain by as much as 12 km of sedimentary strata deposited during, after, and possibly even before periods of
rifting in the late Permian–Early Triassic (rift phase 1) and Middle Jurassic–Early Cretaceous (rift phase 2) (e.g. Whipp et
al., 2014; Bell et al., 2014; Maystrenko et al., 2017). The extension direction of these two phases has long been debated.
Whereas most studies agree that the late Permian–Early Triassic rifting was driven by E-W extension (e.g. Færseth et al.,
1997; Torsvik et al., 1997), Middle Jurassic–Early Cretaceous rifting has been associated with both E-W (e.g. Bartholomew
et al., 1993; Brun and Tron, 1993) and NW-SE extension (e.g. Færseth, 1996; Doré et al., 1997; Færseth et al., 1997) (Fig.
1B). This debate is further complicated by the fact that some of the largest normal faults on the Horda Platform developed
during rift phase 1, but were subsequently reactivated during rift phase 2 (e.g. Whipp et al., 2014; Bell et al., 2014). The
crystalline basement underlying the sedimentary strata formed by terrane accretion during the Sveconorwegian (1140–900





Ma) and Caledonian (460–400 Ma) orogenies (Bingen et al., 2008). Several studies argue that this structural template, in particular the ductile shear zones, controlled the location, strike, and overall pattern of rift-related faulting in the overlying sedimentary successions being reactivated as normal faults, or by limiting the along-strike propagation of faults (e.g. Fazlikhani et al., 2017; Phillips et al., 2019; Osagiede et al., 2020; Wiest et al., 2020).

## 3 Data & Methods

### 3.1 3-D seismic reflection data

In this study, we use one of the largest offshore 3-D seismic data sets ever acquired, which images a large part of the northern North Sea rift across an area of 35,410 km2, and with excellent depth-imaging down to 22 km (i.e., the middle-to-lower crust) (Figs. 1, 2A, 3). The data set was acquired using eight, up to 8-km-long streamers that were towed ~40 m below 75 the water surface. The broadseis technology used for recording covers a wide range of frequencies (2.5-155 Hz), providing high-resolution depth imaging. The data were binned at 12.5 × 18.75 m, with a vertical sample rate of 4 ms. The data was 3-D true amplitude prestack depth-migrated. The seismic volume was zero-phase processed with SEG normal polarity; i.e., a positive reflection (white) corresponds to an acoustic impedance (density × velocity) increase with depth. More details on data acquisition and pre-processing steps are provided by Wrona et al., (2019, 2021a).

### 3.2 Deep learning

Deep learning describes a set of algorithms and models, which learn to perform a specific task (e.g. fault interpretation) on a given data set without explicit feature engineering (e.g. the calculation and calibration of seismic attributes, such as coherence or variance). Deep learning allows the derivation of a fault score volume that highlights normal faults within a 3-D seismic volume. This approach requires that a large number of examples of faults and unfaulted strata are labelled in the 85 training seismic data. We extract 80,000 such examples (2-D squares of 128×128 pixels) from 22 interpreted seismic sections oriented perpendicular to the N-trending rift (Figs. 1A, 2). Note that these seismic sections only constitute <0.1% of the entire 3-D seismic volume. Next, we split these examples into three groups; one set for training (80%), one for validation (10%), and one for testing (10%). We use the first of these to train a deep convolutional neural network (U-Net) designed to perform image segmentation tasks (Ronneberger et al., 2015). Using the validation set, we track the accuracy of the model 90 during training and stop once the loss does not decrease further. Finally, we apply the model to the entire 3-D seismic volume to derive a fault score volume (Figs. 3, 4), an attribute, which ranges from 0 (no fault) to 1 (fault). All details of the workflow and the code are provided by Wrona et al. (2021b, 2021a).






### 3.3 Automated fault network extraction and analysis

Extracting a fault network from the 3-D volume allows us to perform a comprehensive geometric analysis of the fault system using our fault analysis toolbox - fatbox (Wrona et al., 2022). The basic idea is to describe a fault system in 2-D as a network (or graph), i.e. sets of nodes and edges (Fig. 5). Each node marks a location along the fault and each edge connects two nodes. All nodes connected to one another by edges are labelled as a (connected) component.


Our fault extraction workflow consists of these eight steps: (1) extract horizon, (2) Gaussian blur filter, (3) thresholding, (4) cleaning, (5) skeletonization, (6) connect components, (7) add nodes to graph, (8) add edges to graph and (9) split junctions. While applying it to our North Sea target region, we first attempt to capture as many faults as possible by extracting the fault score along a horizon 500 m below Base Cretaceous Unconformity (BCU) (Fig. 1C). Here we observe a

large number of faults, which were either formed in the second rift phase, or formed in the first rift phase and reactivated in the second rift phase (Figs. 4, 6A). Second, we apply a Gaussian blur filter to increase fault continuity (Fig. 6B). Third, we apply a threshold of 0.35 to separate the faults from the background in the fault likelihood (Fig. 6C). This threshold is a tradeoff, which balances capturing as many faults as possible (lower values) and identifying clearly resolvable faults (high values). Four, we further restrict this threshold and essentially filter these points by removing areas smaller than 25 pixels

(Fig. 6D). Five, we collapse the faults to one-pixel wide lines using skeletonization (Guo and Hall, 1992) (Fig. 6E). Six, we label adjacent pixels in the image as connected components (Wu et al., 2009) (Fig. 6F). Each component consists of pixels which are connected to each other. These components represent the faults in the network. At this point, we can build our graph using these connected components of the image (Fig. 6F). Each pixel belonging to a component becomes a node whereas edges are created between neighbouring nodes (Fig 6G-I). This process results in a number of faults with splays,

junctions or intersections being grouped into one connected component (Fig. 7A). To correct this, we split up junctions (nodes with three edges) based on the similarity of strike, i.e. aligned branches remain connected (Fig. 7B,C). This final network is compared to the Base Late Jurassic horizon mapped by Tillmans et al., (2021) (Fig. 8). Additionally, we perform the exact same workflow on ten slices through the fault score volume (1-10 km depth) to capture 3-D fault geometries with depth (Fig. 9).

After extracting the fault system, we calculate a series of typical fault properties using our fault analysis toolbox - fatbox (Wrona et al., 2022) (Fig. 10). First, we calculate the fault length as the sum of the edge lengths of each component (Fig. 10B). Second, we calculate the strike along the fault from neighbouring nodes (Fig. 10C). If we were to calculate the overall fault strike, we would overlook along-strike variations in strike. If we were to calculate the strike as the orientation of each edge, we would only obtain values of 0, 45 or 90°, because the nodes are closely spaced. Instead, we calculate the strike

from the 3rd degree neighbouring nodes (i.e. neighbours of neighbours of neighbours). This assures a robust, high resolution fault strike calculation. Combining the fault length and strike, we can generate a length-weighted Rose diagram (Fig. 10C). Finally, we calculate the fault density as the fault length per area (Fig. 10D).





### 3.4 Comparison to conventional seismic interpretation

We can ask ourselves, "how good are our results compared to a state-of-the-art fault interpretation from the same data set using conventional fault mapping techniques?" (Fig. 8). Tillmans et al., (2021) map the Base Late Jurassic (base of syn-rift sediments associated with rift phase 2) on the eastern flank of the North Viking Graben (see Figs. 1A, 4 for location) using a combination of manual picking and auto-tracking. This horizon is calibrated with 40 exploration wells, which provide direct constraints on the depth of the surface. Tillmans et al. (2021) highlight the fault system by computing the variance attribute

(Chopra and Marfurt, 2007) along the horizon (Fig. 7A). On top of the horizon, we plot the fault network mapped from the fault score extracted 500 m below the easily-mappable Base Cretaceous Unconformity (BCU) (Fig. 8B). This visual comparison shows that while we are missing a few faults in the southwest of the map, we are able to identify and accurately represent most of the faults identified by Tillmans et al. (2021). The missing faults are either overlooked by our model (i.e. false negatives) or result from the difference in the horizons that we compare: Base Cretaceous Unconformity (our study)

versus Base Late Jurassic (Tillmans et al., 2021).

### 4 Observations

Our fault extraction allows us to map a complex network consisting of 7983 individual faults across an approximately 161 km-wide and 266 km-long area, covering 35,410 km2 of the northern North Sea rift (Fig. 7C).

### 4.1 Fault length

Faults vary in length by 3 orders of magnitude - from 50 m to 75.9 km, with some of the longest faults (>30 km) extending from the Stord Basin and Bjørgvin Arch in the south to the Uer and Lomre Terrace in the north (Fig. 10B). In cross-section, these faults have up to several kilometres of displacement and bound rotated half-graben (e.g. Whipp et al., 2014; Bell et al., 2014) (Fig. 3B,C). While we observe some long (up to 20 km) faults in the Viking Graben and Tampen Spur, most faults (>90%) are closely spaced (< 5 km) and relatively short (<10 km long) (Fig. 10B).

### 4.2 Fault strikes

In map view, we observe a complex network consisting of a large number of variably trending faults that display a broad range of intersection styles (e.g., oblique, perpendicular). These faults show a large range of strikes, varying from NW-SE to NE-SW (Figs. 9, 10C). The length-weighted rose plot shows that most faults strike NW-SE (light blue) or NNE-SSW (light orange), with a large number showing intervening strike directions (Fig. 10C). This general divide occurs between

predominantly NW-SE-striking faults along the eastern part of the rift and NE-SW-striking faults in the central and northwestern part of the rift. This divide becomes most evident when comparing faults on the Lomre Terrace (NE-SW) to the adjacent Bjørgvin Arch (NW-SE), at least at the structural level of the Base Cretaceous Unconformity (Fig. 10C).



### 4.3 Fault density

In map view, we observe large variations in fault density 500 below the BCU (Fig. 10D). While dense networks of
intersecting faults result in high density areas (e.g. Lomre Terrace, Bjørgvin Arch) we observe low densities in the Viking
and Sogn Graben, where faults occur at greater depths (e.g. Fig. 9C).

### 4.4 Vertical continuity

The faults extracted at different depths are variable in their vertical continuity (i.e., fault height; Fig. 8). Whereas some
faults, in particular in the Stord Basin, the Tampen Spur, and the Magnus Basin show parallel fault traces from 1 to 10 km
depth (Fig. 9A), we also observe a large number of faults that occur only at shallower (1-5 km) or at greater depths (6-10
km) (Fig. 9B, C). Upon closer inspection, we observe that the faults, which occur continuously between 1-10 km depth, e.g.
in the eastern Stord Basin and the Bjørgvin Arch, are typically large-displacement normal faults with tens of kilometres
spacing (e.g. Fig. 3B, C), whereas the other faults, which only occur between 6-10 km depth (e.g. northwestern Stord Basin),
are usually shorter and more closely spaced (a few kilometres) (e.g. Fig. 9C).

## 5 Discussion

### 5.1 Advantages of deep learning based fault interpretation

When comparing our results to conventional interpretation methods, we can ask ourselves "what value does deep learning
add?". Here, we highlight the advantages of the supervised deep learning-based fault interpretation workflow which we
present in this study. First, we can predict faults in a seismic section in a fraction of the time (5 seconds) required by expert
interpreters (~10 minutes). These differences accumulate, in particular when interpreting such a large data set with >22000
inlines. A conventional fault interpretation of such a large data set can take several months, whereas a trained convolutional
neural network can identify faults across the entire volume within a day on a single GPU (GeForce GTX 1080 Ti).

Second, after identifying faults in seismic reflection data, they also need to be mapped before we can perform the
relevant fault analysis. Here we map the fault network using a series of tools from computer vision and network analysis
compiled in our fault analysis toolbox - fatbox (Wrona et al., 2022) (Figs. 6, 7). Our automated workflow extracts the fault
network in less than five minutes compared to the several weeks to months that would have been required to manually map
the faults in this large data set.  Furthermore, once extracted, we can immediately conduct a number of typical fault analyses
using predefined functions implemented in fatbox (Wrona et al., 2022) (e.g. Fig. 10).

Third, conventional fault interpretations are often binary (fault vs. no fault), but deep learning delivers a score
ranging from 0 (no fault) to 1 (fault), which allows a subsequent analysis of how likely it is to encounter a fault at any point.
This type of analysis is particularly useful for assessing the sealing potential of certain layers for $CO_2$ storage and for
predicting fluid flow during geothermal exploration. Fourth, seismic interpreters typically focus on the largest faults,



whereas our model performs the same prediction across the entire data set and does not differentiate between faults based on their size, shape or orientation. Fifth, given the same data, labels, model and training, our model and results are fully

reproducible, which is not the case for conventional fault interpretations, where the interpreter has to make a myriad of decisions in the process of mapping a fault network.

### 5.2 Complex fault system in the northern North Sea

Our study shows how to reveal the complex geometry of normal fault systems in 3-D seismic reflection data using a combination of deep learning and automated fault extraction. We were able to map an intricate network consisting of almost

8000 individual faults that cover an area approximately 161 km wide and 266 km long (e.g. Figs. 4, 6, 10). This fault network shows large variations in fault length, strike and density, with extremely complex splays, junctions and intersections between these faults (Figs. 7-11). As such, our work goes far beyond typical seismic interpretations in previous case studies, which covered only a fraction of the rift (e.g. Duffy et al., 2015; Deng et al., 2017; Tillmans et al., 2021), or regional studies that mapped <100 of the largest faults using primarily sparse, 2-D seismic sections (e.g. Fig. 1B; Fazlikhani et al., 2017;

Phillips et al., 2019).

### 5.3 Uncertainties during fault mapping

While there are several advantages to our approach, it is worth remembering the uncertainties associated with mapping faults in seismic reflection data. First, seismic reflection data can only image faults with displacement above the seismic resolution (and level of noise) of the data set. The seismic resolution of our data set decreases from 15 m (vertical) and 30 m (lateral)

around 3 km depth down to 180 m (vertical) around 20 km depth (see Wrona et al., 2019; Tillman et al., 2021). Second, the labels we use to train our model are derived from 22 interpreted seismic sections, which, like any seismic interpretation, contains the expertise and biases of the interpreter (e.g. Bond et al., 2007, Bond 2015). Third, the convolutional neural network that we trained achieves an accuracy of 83%, implying that 17% of the data is misclassified (see Wrona et al., 2021). A closer inspection reveals that 36% are false positives (i.e. faults that were overlooked) and 5% are false negatives

(i.e. faults that were misinterpreted) (see Wrona et al., 2021). Despite these limitations, the robustness of our approach is evident when considering along-strike fault continuity across a large number of different seismic lines (Fig. 10, 11).

### 5.4 Future research on automated fault mapping

Based on our work, we can identify three areas for future research on this subject. First, conventional neural networks predict a score from 0 to 1, which we can use as a proxy for how likely it is to encounter a fault at a point. Bayesian neural networks,

on the other hand, allow the prediction of true fault probabilities (e.g. Mosser et al., 2020). Predicting fault probabilities in regional seismic data sets could significantly accelerate the screening for and risk assessment of potential $CO_2$ storage sites (see Wrona and Pan, 2021). Second, in addition to predicting where faults occur, we can explore the prediction of other fault

properties, such as displacement, fault zone permeability or even the time when they were active. This would significantly allow us to study the spatial and temporal evolution of fault systems in high resolution at a regional scale. Third, while our

fault extraction workflow currently focuses on mapping fault networks in a series of 2-D slices or horizons, we really need freely-available methods to generate 3-D fault surfaces, which allow for complex fault splays, junctions and intersections, as observed here.

## 6 Conclusions

This study shows that the combination of deep learning and network analysis applied to 3-D seismic reflection data allows

us, for the first time, to map almost 8000 normal faults across the entire northern North Sea rift. These faults form an intricate network with complex relationships (e.g. splays, junctions, intersections) including large variations in fault length (50 m to 75.9 km) and strikes (NW-SE to NE-SW). As such, this work goes far beyond previous seismic studies by providing high resolution fault maps at a regional scale in a fraction of the time required by conventional interpretation methods.

**Acknowledgements**

We would like to thank The Norwegian Academy of Science and Letters (VISTA), The University of Bergen and The Initiative and Networking Fund of the Helmholtz Association through the project "Advanced Earth System Modelling Capacity (ESM), The Geo.X Network and Deutsche Forschungsgemeinschaft (Projektnummer 460760884) for supporting this research. S. Brune has been funded through the Helmholtz Young Investigators Group CRYSTALS (VH-NG-1132). I.

Pan acknowledges the NUAcT fellowship for partially supporting the work. We are grateful to CGG, in particular Stein Åsheim and Jaswinder Mann, for the permission to present this data and publish this work. We thank Schlumberger for providing the software Petrel 2019© and Leo Zijerveld for IT support.

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








**Figure 1: A** Structural overview map of the northern North Sea basin system (from Tillmans et al., 2021 after Færseth, 1996). Blue rectangle marks the outline of the seismic survey in this study. ESB = East Shetland Basin, B-S = Brent-Statfjord Fault, G-V = Gullfaks-Visund Fault, MS = Måløy Slope, HP = Horda Platform. **B** The base rift surface (base Permo-Triassic rifting) time-structure map in the northern North Sea rift (from Fazlikhani et al., 2017) and the geology of southwestern Norway, showing the
general onshore and offshore structural configuration in the study area. Bold black lines highlight major rift-related normal faults displacing the base rift surface where all units older than Upper Permian are considered basement. Black lines in the background show some of the 2-D seismic reflection surveys used by Fazlikhani et al. (2017). NSDZ, Nordfjord-Sogn Detachment Zone; BASZ, Bergen Arc Shear Zone; WGR, Western Gneiss Region; ØC, Øygarden Complex (gneiss); ØFS, Øygarden Fault System; HSZ, KSZ, and; SSZ: Hardangerfjord, Karmøy, and Stavanger shear zones, respectively. **C** Regional interpretation of the structure of
the northern North Sea after Færseth (1996).





**Figure 2: A** Example seismic section across the northern North Sea. Amplitudes are scaled for machine learning **B** Example of
fault interpretation of the section used to train a deep convolutional neural network for fault prediction.








**Figure 3: Examples of seismic sections extracted from fault score volume of the 3-D seismic data set. Note that these sections were not part of the training data, but are actually 6.25 km away from the closest interpreted seismic section (see Fig. 1A).**






**Figure 4: Surface capturing tectonic faults extracted from fault likelihood volume. The surface was extracted 500 m below the Base Cretaceous Unconformity, where we observe a large number of faults, which were either formed or reactivated in the second rift phase. White rectangle shows the area used for validation (Fig. 8) and the red rectangle indicates the area where we demonstrate our fault network extraction workflow (Fig. 6).**



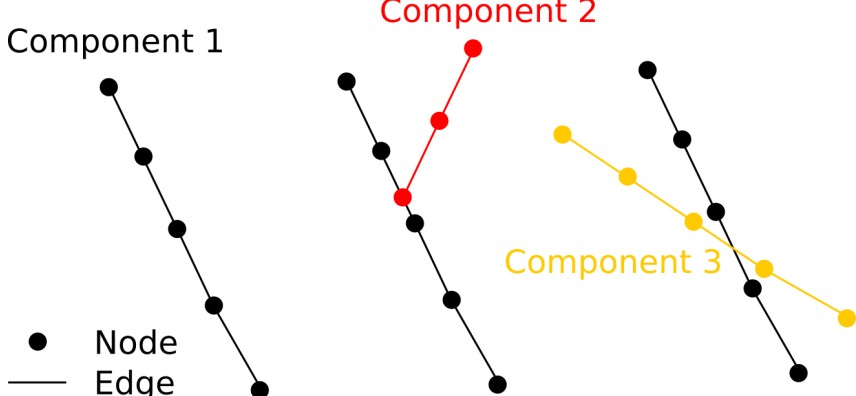

**Figure 5: Schematic illustration of fault network (or graph) with nodes, edges and components. Each node marks a location along the fault. Each edge connects two nodes and each (connected) component indicates all nodes connected to one another by edges.**







**Figure 6: Fault network extraction workflow showing: A Fault score extracted along the surface (500 m below BCU). B Gaussian Blur filter (σ=2) of surface. C Threshold (0.35) of filter. D Cleaned threshold where small patches are removed. E Skeleton of cleaned threshold. F Connected components of skeleton. G Network nodes based on components. H Network edges based on components. I Network nodes and edges combined. Note that colours in F, G and I indicate connected components (i.e. individual faults), before splitting (see Fig. 6).**

430

435



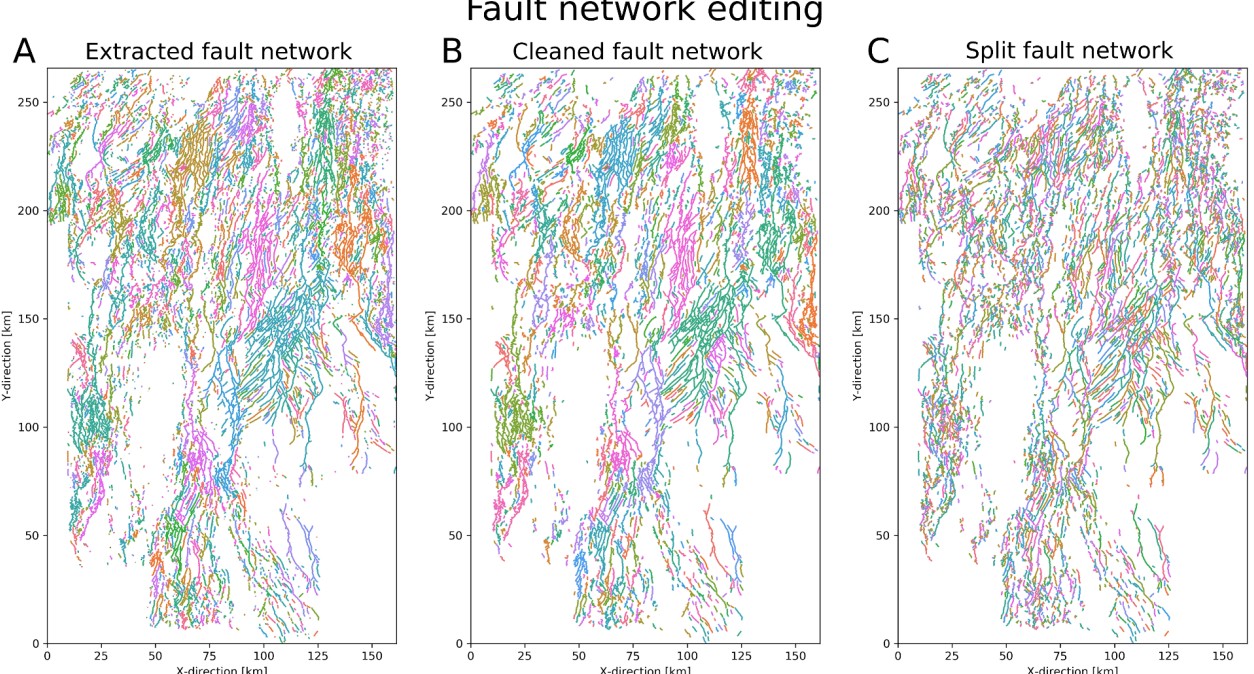

**Figure 7: A** Fault network extracted from BCU (Fig. 4D). Note the large areas with the same colours resulting from multiple faults being grouped into one connected component **B** Fault network after removal of noise (i.e. small components). **C** Fault network after splitting junctions previously connecting splaying and intersecting faults. Note that large connected components are split up and individual faults are highlighted by different colours.



## Time-structure map   + extracted fault network

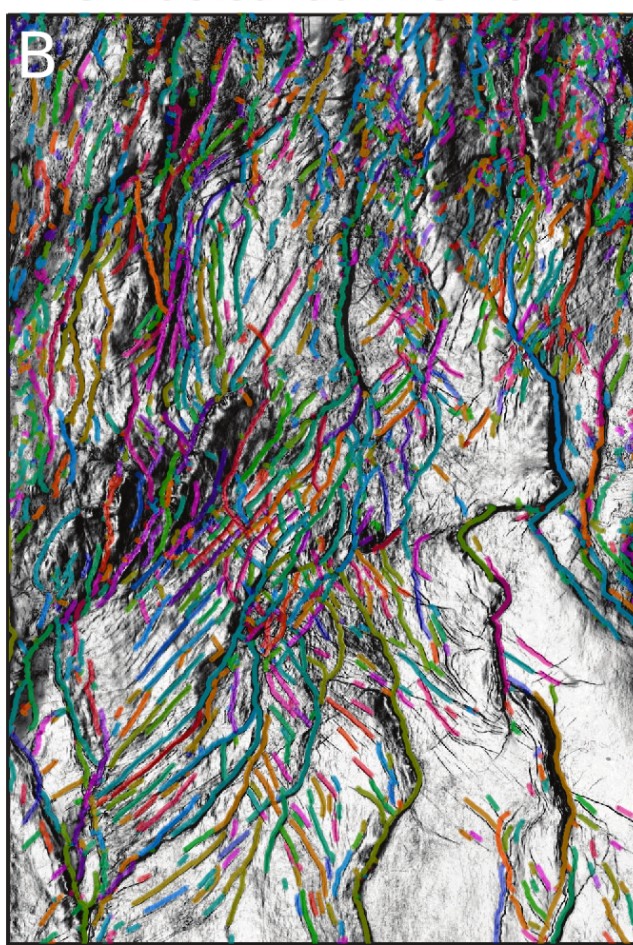

**Figure 8: Comparison of A Base Late Jurassic time-structure map interpreted by Tillmans et al., (2021) and B Automatically-extracted fault network 500 m below Base Cretaceous Unconformity. Faults are distinguished by colour.**







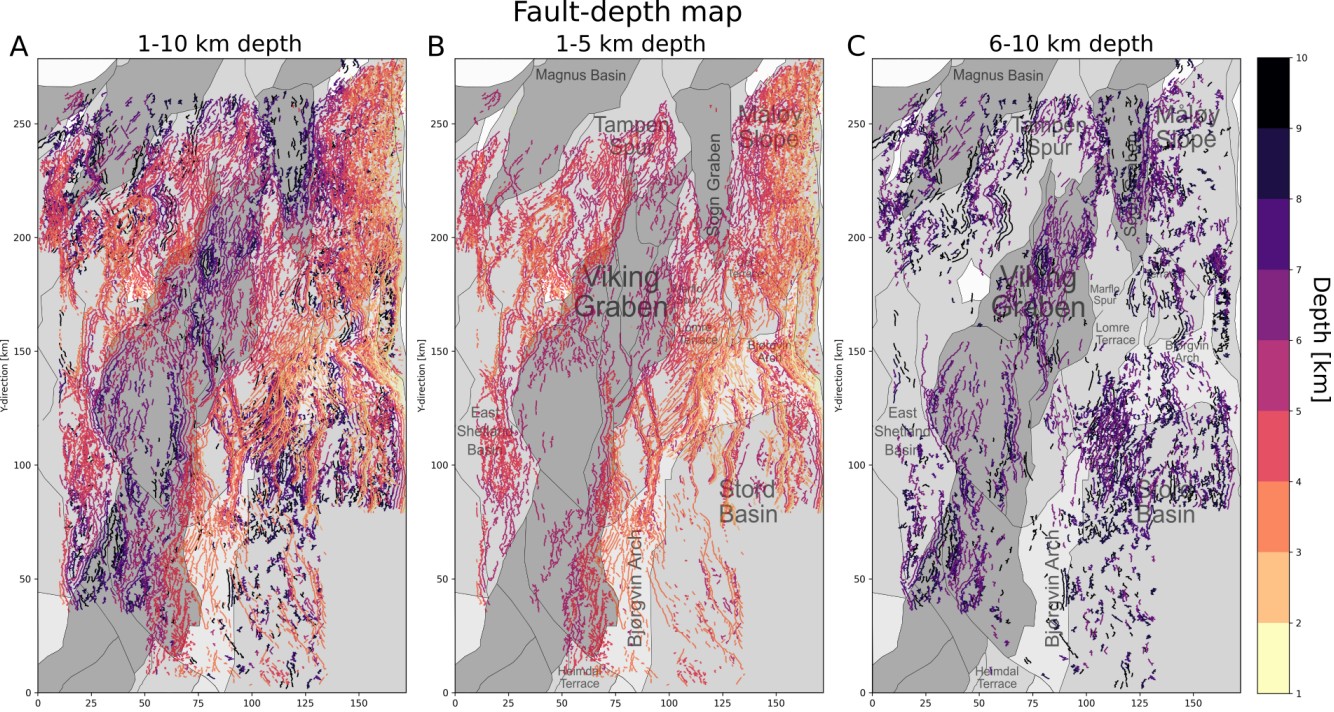

**Figure 9: Fault map of the northern North Sea extracted every kilometre between 1-10 km depth (A), 1-5 km depth (B) and 6-10 km depth (C) with structural elements from the Norwegian Petroleum Directorate or NPD (2022).**










**Figure 10: A** Structural elements of the northern North Sea Rift (NPD, 2022) **B** Fault lengths (500 m below BCU) on top of structural elements. **C** Fault strikes (500 m below BCU) on top of structural elements with length-weighted Rose diagram. **D** Fault density on top of structural elements.





**Figure 11: 3-D perspective of the northern North Sea rift showing the Base Cretaceous Unconformity overlain with faults (black) extracted from 3-D seismic reflection data with deep learning. Vertical exaggeration of 5.**