# Peer review of "Complex fault system revealed from 3-D seismic reflection data with deep learning and fault network analysis"

_EGUsphere, 2022_

## Author Comment (AC2)

… In summary, I do not see any major issues with the manuscript and it should therefore be accepted subject to minor revisions and comments.

Line 29: Would the same study be possible for other types of faults? Have only normal faults been labelled in the data labelling process for training the deep neural network? Would the network be able to distinguish normal faults from reverse faults for example?

> Yes, similar studies should be possible for different fault types if they are resolved by the seismic dataset. Strike-slip faults, for example, are notoriously difficult to resolve with seismic reflection data, as they often show little to no vertical offset of reflectors. Normal and reverse faults on the other hand often show these offsets, which neural networks can learn to recognize. Generally, machine learning models should be able to identify faults based on the seismic signature of the damaged rocks (e.g. lower amplitudes, more chaos), so if these faults show an expression in the seismic dataset (i.e. we can recognize them), we should be able to train a model to recognize them, too.

> In our study, we labelled all major faults in the training data, which are primarily normal faults (probably >99%). A handful of these normal faults may show evidence of minor inversion, but they all remain in net-extension, i.e., the hanging wall has moved down relative to the footwall.

> Our current model has not been trained to and is thus unable to distinguish between normal and reverse faults. Moreover, it is certainly a challenging task to differentiate between these two fault types, because the model would need to be able to recognize and correlate reflectors on either side of the fault to define the sense of displacement.

Line 46 and 47: I feel the wording around "proof of concept" and "has yet to" could be improved. There is a lot of groundwork that needs to be done to understand the ability of these deep neural networks to become a reliable source of knowledge. Just because there has not been a study to evaluate the insights gained does not mean there wasn't potential to do so coming from numerous works where the ability of the networks to detect faults have been established, some of which have even been published including Model weights e.g. Wu et. al.

> We changed the wording to (L51-55 in the revised manuscript):

> *'Supervised deep learning allows us to detect and map faults in seismic reflection data (e.g. Wu et al., 2019; Mosser et al., 2020; Wrona et al., 2021a), but until now many of these studies have laid the foundation by focusing on detecting faults rather than studying the geometry of these faults.'*

Line 50: I am not sure I like the use of "<0.1% of data volume". I have made this statement myself before, but I believe rather than focusing on the reduced volume we should focus on data quality. Apart from the fact that seismic data have strong lateral correlations making additional neighbouring data less diverse, we can also consider other criteria: How many of the relevant types of faults have been mapped, and how many noise modalities have been incorporated? Nothing to change here for now, but potentially something to address in the future?

> Yes, these are very good points, which cannot at this stage, but are definitely worth considering at the start of future projects. Moreover, we could incorporate

uncertainty estimates and active learning to minimise the number of faults labelled to the most important (i.e. uncertain) areas.

Line 88: Accuracy is a metric that is not well suited for class imbalanced problems such as fault detection. Have you considered using the F1 score?

Yes, we have used several other metrics, such as F1 score and IOU, in this and previous studies (e.g. Wrona et al., 2021a), but decided to use accuracy here, because most readers probably have a natural feeling for its meaning.

Line 89: Did you monitor the training or validation loss?

The validation loss. We clarified this in the revised manuscript.

Line 93: Have you considered also publishing the weights?

We did consider this, but unfortunately we are not allowed to publish the weights due to confidentiality.

Line 108: The faults identified at threshold < 0.3 may be small faults but also misclassifications, the same goes for larger faults where the threshold may be >0.3.

How did you determine an appropriate threshold, given that you also filter small faults during the extraction phase?

This threshold is not set in stone and very much up to the user to choose appropriately to the task at hand. In our case, we chose the threshold to pick up as many faults as possible without distorting the geologically plausible geometry of the large faults (L158-161). Because some of the smallest 'faults' are probably local misclassifications, we removed them during subsequent filtering.

Ideally, this threshold would be tied to a physical fault property (e.g. fault size or displacement) to communicate which faults are picked up and which are not.

Line 174: Should we not add the training time, model validation and QC, and labelling time as well to make the comparison fair? In this case the model was created specifically for this dataset, so the cost would not amortize over the application to many other datasets.

We have added this sentence to the manuscript (L269-271):

*Note that this comparison does not include the time required to label the training data (~2 days), train the initial model (~4 hours), and fine-tune and select the final model (days-months).*

Line 180: The fatbox toolbox was mentioned earlier, do you need to repeat it?

We have removed the reference to fatbox at this point of the paper.

Line 185: It is mentioned later that the fault score should not be equated with what I assume is a calibrated fault probability, yet here you mention that it is possible to determine how likely a fault is to occur. From a miscalibrated model this judgement can be misrepresented. See

Runhai Feng, Dario Grana, and Niels Balling, (2021), "Uncertainty quantification in fault detection using convolutional neural networks," GEOPHYSICS 86: M41-M48. https://doi.org/10.1190/geo2020-0424.1

Lukas Mosser and Ehsan Zabihi Naeini, (2022), "A comprehensive study of calibration and uncertainty quantification for Bayesian convolutional neural networks — An application to seismic data," GEOPHYSICS 87: IM157-IM176. https://doi.org/10.1190/geo2021-0318.1

for examples of how this can be ameliorated. Including validations against synthetic data and corresponding metrics (Mosser & Naeini 2022).

> Yes, we have rewritten this section to focus on the qualitative selection of faults (L279-284):
>
> *Third, conventional fault interpretations are often binary (fault vs. no fault). In contrast, deep learning delivers a fault likelihood score ranging from 0 (no fault) to 1 (fault). Although this score is not a true measure of fault probability (see discussion by Mosser and Naeini, 2022), it nevertheless correlates with fault visibility (i.e. faults, which are well-resolved by the data, are associated with higher fault scores). This allows users to qualitatively select the faults that they want to analyze using a score threshold (as done herein).*
>
> The uncertainty quantification is discussed in the comment on Line 214.

Line 188: Wouldn't you have to determine this empirically if your model does not differentiate based on strike, shape, or size? Some indications of this are the inability to detect faults that are oblique or parallel to inline crossline detection. In this case this is a given by design since the model is a 2D network. Validations using synthetic fault geometries would certainly help support or validate such assumptions.

> Yes, this is a good point. We have rewritten this statement (L288-289):
>
> *'Fourth, seismic interpreters typically focus on the largest faults, whereas our model performs the same prediction across the entire data set irrespective of the size of the faults encountered.'*
>
> Our training data consists of E-W trending seismic sections oriented perpendicular to the rift-axis (N-S), which means that faults aligned with these sections were probably not labelled and as a result, the model has not been able to learn to recognize them. We thus cannot assume that the model detects all faults regardless of their strike. Interestingly, the model shows a remarkable ability to map faults with obliquities of up to 60° (e.g. Fig. 10C). Nonetheless, we removed the statement regarding the invariance of the model to fault orientation from the manuscript for the reason outlined above.

Line 214: The statement "we can use (the fault score) as a proxy for how likely it is to encounter a fault" and the subsequent statement of "Bayesian neural networks … able to predict true fault probabilities" are a bit contradictory. Would you agree that using the fault score as a proxy can only be done if the corresponding scores can be calibrated against independent data? I am not sure how you see the fault score being used in a

quantitative manner, or whether you mean that the fault score can be used in a qualitative manner to indicate the presence of a fault which should not be mistaken for a true probability. Examples of such approaches are highlighted e.g. in Mosser & Naeini 2022 showing miscalibrations of U-Nets trained with balanced loss functions.

> Yes, a very good point. We have rewritten the sentence (L318-321):
>
> *Based on our work, we can identify three related areas for future research. First, conventional neural networks predict a fault score from 0 to 1, which seems to correspond to the visibility of the fault in the dataset.*

Line 221: Agreed, 3D fault extraction libraries would make for a great addition to the open-source software domain.

> Yes, that would be super useful for observational datasets, such as 3D seismics, as well as for large 3-D analogue and numerical models.

Regarding Figure 3, could the authors address why they choose to highlight only faults with a probability > 0.5 in the colour map, as opposed to figure 4?

> Figure 4 shows the whole range of values of the fault score (0-1). In Figure 3, we wanted to show the correspondence between the seismic data and the fault score. For the overlay, we need to define a cutoff value (0.5) below which the fault score becomes transparent, so that the seismic data is visible.

Do the authors have any recommendations on how to choose filter sizes for the Gaussian blur? Is there a reason behind using a Gaussian blur to preprocess the fault score maps? Why would thresholding not be sufficient?

> Using thresholding only, we can generate many aligned but disconnected fault segments, each labelled as an individual fault. This is problematic when calculating fault length. Smoothing the fault score prior to thresholding increases the lateral fault continuity allowing us to extract long, geologically plausible faults, geometrically consistent with those mapped in other studies of the same study area (see Fig. 8, Tillmans et al., 2021).
>
> We would recommend using a small filter, so that smoothing only applies locally and does not connect distant faults with one another.

In figure 8, we can clearly identify some major faults not picked by the network and extraction in the lower right corner of the image. Is the seismic data that was used the same? It is addressed generally in the text, but I couldn't see if it was the same seismic dataset.

> It is the same seismic dataset. This is now clarified in the manuscript (L189) and the related figure caption (L201).

Have the authors considered processing the dataset in the main fault strike orientations i.e. NE-SW and NW-SE instead of inline direction? This could help better identification of oblique faults which are otherwise not well-imaged.

> This certainly makes sense for future studies. We have spent a good month splitting up the dataset, exporting it from Petrel, predicting the fault score in

Python and merging predicted subvolumes again in Petrel. There is a plugin from Cegal which allows an easier transfer and processing of the data now.

How was the Fault density square area measured, and how was the size of the averaging element chosen for Figure 10 D?

The fault density was measured as fault length per square area. The squares have an edge length of ~3.6 km, which was chosen for visual purposes.

---

## Author Response (AR2)

Dear Dr. Malinowski,

thank you very much for the quick turnaround and these helpful comments. We have addressed them below and in the revised version of the manuscript. Following your suggestion, we included many of the points from the previous reply letter in the manuscript to give the reader a bigger picture. Below you can find a point by point response (green) to your suggestions giving line numbers corresponding to the tracked changes version of the manuscript. We hope that you will find the manuscript suitable for publication now and would like to thank you again for your time and effort,

Best regards,

Thilo Wrona et al.

**REVIEW**

The revisions prepared do not completely address the concerns of the reviewers in all cases. I would suggest that for

1) Reviewer 1 - line 29 comment - the authors add some of this information about limitations or what is known into the manuscript, as this question will likely be of interest to many readers.

> This has been added to line 339-347 in the revised manuscript with tracked changes.

2) Reviewer 1 - Line 50 - Again, this would be something for the authors to add a few notes on in the discussion or conclusion

> This has been added to line 359-362 in the revised manuscript.

3) Reviewer 1 Line 88- As many methods use different scores, it would be ideal to add these in an appendix, or make a quick note of them. Particularly, the F1 score is quite common.

> This has been added to line 130

4) Line 93- what is confidential about the weights? is this a commercial software or algorithm? Perhaps weight ranges could be mentioned.

> The weights are the machine learning model. They were generated using the confidential 3-D seismic data set and are thus a derivative, which we are not allowed to publish.

5) Reviewer 1 - line line 221 and Figure 3 comments - again, ideal to add these thoughts to the manuscript, as well as following comment on filter sizes.

> Yes, this has been added to the manuscript (L178-180, L368-369) and the figure caption (L137-139).

6) Reviewer 1 - final comment. Please add this information to the manuscript.

> This has been aded to line 232-234

7) Reviewer 2 - It would be ideal to further address this comment by adding more of the geological insights and improvements into the manuscript discussion, to highlight the advancements that this method allows - specific examples (perhaps even adding a geologic comparison figure to highlight) would really help readers see the improvements that can be

obtained in terms of improvements in geologic insights, and perhaps encourage them more to try out this method on their datasets.

> We completely understand the intention of highlighting new geological insights from this study. However, a detailed, in-depth analysis of this fault system will be required to come to concrete conclusions regarding new geological findings (e.g. on the differences between tectonic domains or the influence of pre-existing structures). These fault analyses and geological discussions go beyond the scope of this manuscript, which already covers quite a range of subjects (geophysics, machine learning, image processing, network analysis). We are convinced that the readers can recognize the potential of our approach and further papers on the geological interpretation of this complex fault system will follow.